# Effective and Efficient DDoS Attack Detection Using Deep Learning Algorithm, Multi-Layer Perceptron

**Sheeraz Ahmed** [1], **Zahoor Ali Khan** [2], **Syed Muhammad Mohsin** [3,4,*], **Shahid Latif** [1], **Sheraz Aslam** [5,6,*], **Hana Mujlid** [7], **Muhammad Adil** [1] and **Zeeshan Najam** [8]

1  Department of Computer Science, Iqra National University, Peshawar 25000, Pakistan
2  Faculty of Computer Information Science, Higher Colleges of Technology, Fujairah 4114, United Arab Emirates
3  Department of Computer Science, COMSATS University Islamabad, Islamabad 45550, Pakistan
4  College of Intellectual Novitiates (COIN), Virtual University of Pakistan, Lahore 55150, Pakistan
5  Department of Electrical Engineering, Computer Engineering, and Informatics, Cyprus University of Technology, Limassol 3036, Cyprus
6  Department of Computer Science, Ctl Eurocollege, 3077 Limassol, Cyprus
7  Department of Computer Engineering, Taif University, Taif 11099, Saudi Arabia
8  CEO, Ultimate Engineering Consultants Private Limited, Peshawar 25000, Pakistan
*  Correspondence: syedmmohsin9@yahoo.com (S.M.M.); sheraz.aslam@cut.ac.cy (S.A.)

**Abstract:** Distributed denial of service (DDoS) attacks pose an increasing threat to businesses and government agencies. They harm internet businesses, limit access to information and services, and damage corporate brands. Attackers use application layer DDoS attacks that are not easily detectable because of impersonating authentic users. In this study, we address novel application layer DDoS attacks by analyzing the characteristics of incoming packets, including the size of HTTP frame packets, the number of Internet Protocol (IP) addresses sent, constant mappings of ports, and the number of IP addresses using proxy IP. We analyzed client behavior in public attacks using standard datasets, the CTU-13 dataset, real weblogs (dataset) from our organization, and experimentally created datasets from DDoS attack tools: Slow Lairs, Hulk, Golden Eyes, and Xerex. A multilayer perceptron (MLP), a deep learning algorithm, is used to evaluate the effectiveness of metrics-based attack detection. Simulation results show that the proposed MLP classification algorithm has an efficiency of 98.99% in detecting DDoS attacks. The performance of our proposed technique provided the lowest value of false positives of 2.11% compared to conventional classifiers, i.e., Naïve Bayes, Decision Stump, Logistic Model Tree, Naïve Bayes Updateable, Naïve Bayes Multinomial Text, AdaBoostM1, Attribute Selected Classifier, Iterative Classifier, and OneR.

**Keywords:** DDoS attack; attack; attack detection; botnet; MLP classifier

## 1. Introduction

In today's fast-paced world, where the number of internet-connected devices is increasing and online applications are growing at a rapid pace, information security is becoming an absolute necessity. Since the beginning of the World Wide Web, 1.2 billion websites have been developed [1], and a huge number and variety of online applications are integrated with various web services, such as e-commerce, online banking, online shopping, online education, e-healthcare, and industrial control systems (ICS) for critical infrastructure, etc.

Nowadays, cyber attackers are highly skilled and well-equipped to carry out successful attacks on businesses and governments [2]. Cybercrime is big business today, and the volume of stolen information is enormous. There are many different categories of malware [3]. This poses a huge risk to governments, businesses, and consumers around the world. We do not have to go far back in time to remember the massive attack on a bank in Bangladesh, where USD 81 million was reportedly stolen. This is a constant reminder

of how effective these attacks can be; the bank's own computers were used to transfer large sums of money. No business is safe, no matter how large. Statistics show that 20% of affected businesses fall into the small business category, 33% into the SME category, and 41% into the large business category. The more widespread the threat, the more important it becomes to be aware of the issues and protect the important information. Eighty-two percent of organizations have been exposed to at least one or more attacks in which data are stolen and used to cripple the victim's services. The organizations that were affected by DDoS attacks reported a 26% drop in performance of their services and 41% reported an outage of the affected services [3]. Figure 1 shows an environment of DDoS attacks.

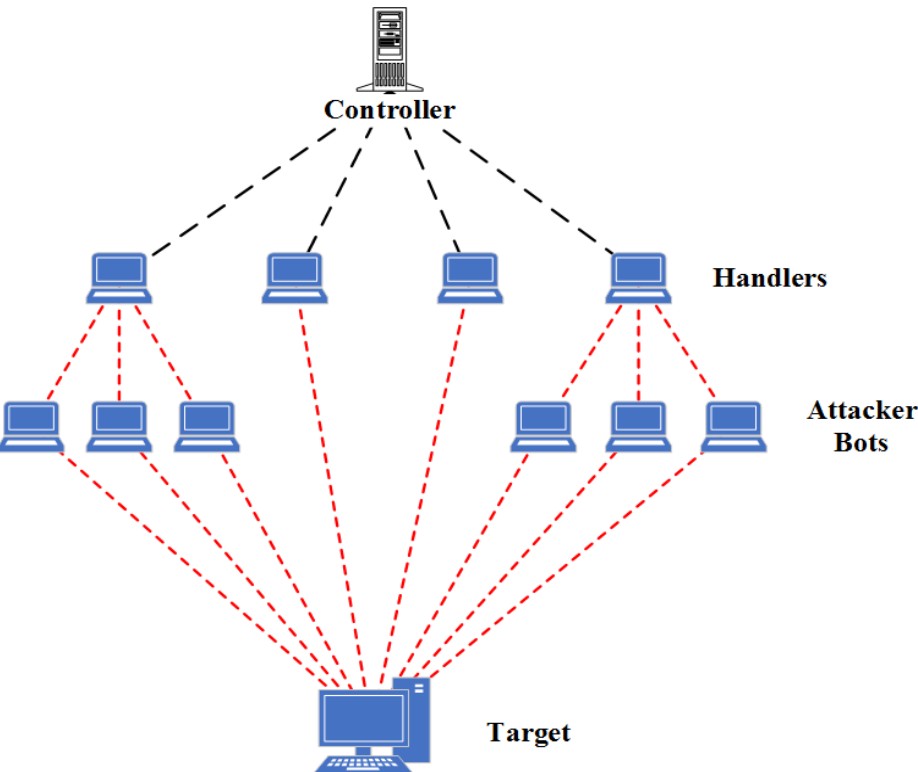

**Figure 1.** DDoS attack environment.

The attacker makes precise imitations of human users' behaviors in an effort to avoid being noticed while conducting the assault. To launch an HTTP-GET flood DDoS assault, the malicious user separates his attack techniques based on rate, admission pattern, etc. Several significant concerns and challenges that have surfaced from recent research have drawn increasing attention in the area of HTTP-GET flood DDoS assault detection. When working in the vicinity of conflicting HTTP-GET flood DDoS attacks [1], numerous challenges arise, that are also only partially addressed or unresolved. DDoS attacks are disciplined, distributed, and remotely organized networks that use deployed computers (also called Bots or Zombies) to send an immense number of uninterrupted and synchronous requests to the victim system(s). DDoS attacks are increasing in strength, regularity, and complexity.

Malicious users are constantly evolving their experience, adapting their techniques, and using advanced technologies to launch various DDoS attacks. While there are various solutions to detect, defend against, or mitigate DDoS attacks, malicious users continue to develop new approaches and means to circumvent these countermeasures [3]. DDoS events are still among the biggest threats to the network. Recently, DDoS outbreaks at the application layer of internet servers have become widespread, resulting in huge revenue losses for targets [4]. In TCP/IP layer attacks, the online server is crushed and the number of requests per second is limited. Slowloris, zero-day attacks, and DDoS assaults that take advantage of Apache or Windows vulnerabilities fall under this category [5].

The solutions offered to understand DDoS attacks at the TCP/IP layer capture only a subset of DDoS incidents at the application layer. The resolutions that detect entire types of application-layer attacks are very complicated in formula. One set of tasks in detecting a DDoS outbreak at the TCP/IP layer is the unavailability of landscapes to detect such incidents [6]. HTTP-GET DDoS attacks are a risk for all web servers, as bots are able to impersonate humans and make it difficult to distinguish malicious requests from real ones. Regardless of industry or scale, enterprises around the world are increasingly becoming targets of DDoS attacks.

Complexity and strength of these attacks are increasing exponentially as the number of admitted systems increase, vulnerabilities go un-patched, and business impact increases [7]. DDoS attacks have a strong impact on the cyber domain. Cyber attacks are feared to disrupt the regular functioning of the organization through IP overflow, bandwidth spoofing, intensive memory resources, and root sane or mouse damage [8]. A slow-moving DDoS attack has the capacity to mimic real traffic with its traffic. It is simple to avoid detection by current systems. Based on their rank values, rank correlation techniques can detect significant differences between attack traffic and legitimate traffic [9].

DoS attack has serious impacts on information servers, internet servers, and cloud computing servers [10,11]. Botnets, DDoS, hacking, malware, pharming, phishing, ransomware, spam, spoofing, and spyware are some of the most frequent hazards [12]. According to Ginni Rometty, Chief Executive Officer IBM, the biggest risk to any or all businesses worldwide is a cyberattack. With that, there is an increase in cybercriminals [9]. Malicious users use numerous hacking methods to hack client servers. DDoS attacks are very wide-ranging attacks and occur between other cyber attacks; detecting DDoS attacks is not easy. Three basic types of DDoS attacks are described below.

### 1.1. Volume Based DDoS Attack

Volume based DDoS attacks consist of faked packet floods such as ICMP floods, UDP floods, and others. The objective of this attack is to use all of the target site's bandwidth, and it is measured in bits per second (bps). Various prominent types of DDoS attacks are shown in Figure 2.

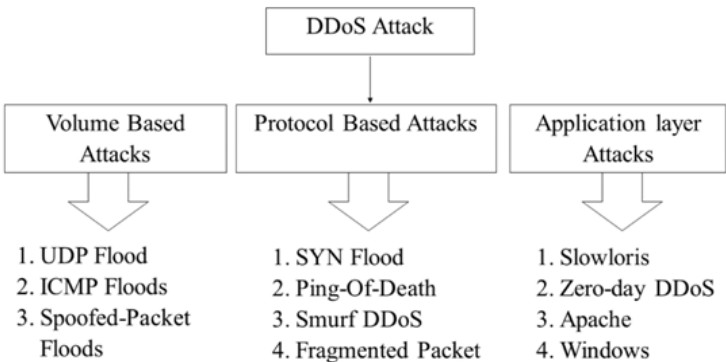

**Figure 2.** Basic types of DDoS attacks.

### 1.2. Protocol Based DDoS Attack

Protocol based DDoS attacks appear in a variety of forms, such as SYN floods, fragmented pack attacks, ping of death, smurf DDoS, and others. Attacks are measured in packets per second (pps). Such types of attacks use real server resources, as well as those of central communications devices like firewalls and load balancers.

### 1.3. TCP/IP Layer Based DDoS Attack

TCP/IP layer based DDoS attack comprise GET/POST floods, low and slow-speed attacks, potential Windows or Open BSD attacks, Apache-driven attacks, and more. Such

attacks seem to be legitimate and innocent applications, and they target the web server. The extent is measured by requests per second.

The number of attacks and the associated traffic volume continue to increase dramatically. With such traffic intensity, the network infrastructure upstream of the intended victim is also severely impacted, so attack traffic must be filtered as close as possible to the sources of attack. However, it is difficult to predict and identify such nodes, as attacks originate from widely distributed nodes and spread across multiple locations. To successfully respond by disrupting traffic, the mitigation approach must detect malicious traffic and respond with minimal impact on legitimate traffic. The attacker launches a new attack, known as increasing DDoS attack and proxy DDoS attack. We develop a detection algorithm to solve this problem. The detection algorithm uses deep-learning techniques to detect malicious traffic and separate legitimate traffic from malicious traffic. The algorithm classifies traffic into three categories: (1) normal traffic (2) suspicious traffic (3) malicious traffic. The main contributions of this study are summarized below.

1.  We addressed novel application layer DDoS attack by analyzing the characteristics of incoming data packets including size of HTTP frame packets, number of IP addresses sent, constant mappings of ports, and number of IP addresses using proxy IP.
2.  We analyzed the client's behavior in public attacks using standard datasets, CTU-13 dataset, real web logs (dataset) from our organization, and experimentally created datasets from DDoS attack tools such as Slow Lairs, Hulk, Golden Eyes, and Xerex.
3.  A deep learning classification algorithm, multilayer perceptron (MLP), is proposed to evaluate the effectiveness of attack detection based on metrics.
4.  Our proposed MLP classification model provided the lowest value of false positives as compared with conventional classifiers such as Naïve Bayes, Decision Stump, Logistic Model Tree, Naïve Bayes Updateable, Naïve Bayes Multinomial Text, AdaBoostM1, Attribute Selected Classifier, Iterative Classifier, and OneR.

The rest of the article is organized as follows. Section 2 briefly describes the literature review; the problem motivation is discussed in Section 3. Chart flow and research methodology are presented in Section 4. The proposed attack classification model is briefly described in Section 5 and simulation results are elaborated upon in Section 6. Finally, Section 7 concludes this study along with future work.

## 2. Literature Review

Machine learning algorithms are being widely used by research community in every field of life. Prominent application areas of machine learning algorithms include image processing, forecasting, recommendation systems, healthcare, banking system, defence, education, robotics, etc. [13]. Deep learning is a subset of machine learning. In this study, we have used a deep learning algorithm, namely, multilayer perceptron (MLP), for effective and efficient detection of DDoS attacks. State-of-art literature on DDoS attack detection is summarized in the following.

Authors of [14] have focused on mitigating multi-page HTTP DDoS attacks with slow-moving targets that target public servers. The conceptual proof model was used in a simple and validated the argument. In [15], the authors compared the probability similarity between cyber attack, DDoS, and mathematical prototypical probability, Levy Walks. This variation aimed to determine the suitability of Levy walk as prototypical similarity with DDoS potential features. In [16], the authors experimented with the clever subject of comedy measurement that utilizes a conference seeking philosophy and a brilliant channel that sets shares in the traditional way. Multilayer perceptron with genetic algorithm (MLP-GA) is proposed in [17] to detect DDoS attacks. The authors examined the areas of incoming pockets as well.

It is assumed that the non-receiver of an unusual collection returns once at the time of publication. However, the authors of [18] provided a sequence of events for experimental distribution to test the capabilities. The authors did not show a positive impact on stock recovery, but in cases where DDoS attacks cause disruptions within the services sent by the

client, the study experienced a strong negative impact. The current unit of current methods was created due to the actual malfunction of DDoS attack detection in the application layer.

The authors of [19] developed a phase-based system with downloading local packets, fine-field extraction of these local units needed for detection, and the use of a separator for attack detection. The study at [20] examined the impact of a DDoS attack on a state-of-the-art gift network and evaluated network security mechanisms such as a router protection system and network servers. In [21], the authors presented a solution for such a type of DDoS attack. When the server exceeded the limit of its application, the author then proposed a solution and sent a random number, which can be selected at an unconsidered time value, to the requesting client.

Research at [22] provided a design that increases resilience to DDoS attacks by upgrading the roles of a virtual network and the software that defines a network. In the first phase, the proposed design defines the roles of the virtual network by solving the linear system. In phase two, to increase the previous protection against DDoS attacks, special VNF filters and a second path through these VNF filters were established by solving another linear system. SDN controller switches routes with a second attack to DDoS traffic filtering methods to prevent congestion under DDoS attacks. In [23], the authors provided a flexible identifier that is set periodically in the background and can make additional data selections. The authors provided applications related to the occurrence of a DDoS-based attack group and a metalfolding model that combines two orthogonal oddity-based attack modes.

In [24], the authors provided a DDoS detection combining a fully based standard and an exceptionally dependent method in which three types of machine applications are found. The author first studied the performance of the proposed system under conditions enforced by normal saturation and TFN2K attacks. Then, the authors apply small costs, such as a saturation period with key traffic attack points, to soak the victim. The authors of [25] investigated our hypotheses about the problem in the existing diagnosis method of the attack on the DoS application base with a strong attack on the algorithm of the CUSUM system. In [26], the researchers developed bio-roused conflicts, based largely on the DDoS Assault framework, with the goal of achieving a faster space. The given prototype can be a bio-roused bat algorithm system, which usually handles the fast and timely location of a DDoS application over HTTP floods.

The authors of [27] proposed a cloud-based firewall to reduce DDoS attacks on the smartest grid network AMI. The Promoted Firewall is not only able to reduce the impact of DDoS attacks, but can also prevent attacks before they start. In [28], the authors demonstrated another planning phase to detect and prevent multiple DDoS TCP (CS_DDoS) attacks during the day. The proposed CS_DDoS framework provides response protection for deleted records. In [29], the authors provided an event detection module to limit the proliferation of internet of things (IoT) services. It was modified from the current monitoring modules with information-based filters. The proposed module focuses on system behavior during DDoS attacks and detects them using NTP-collected information used in the synchronization service. The author performed a demo test with an advanced module that generates a fake DDoS attack. The result showed that the deployed modules obtain high memory and accurate values, which show their effectiveness in capturing real-time events in IoT.

A study done by [30] presented exponentially weighted moving normal (EWMA) search for amazing mine learning and DDoS base discovery attacking internet of things (IoT). The authors investigated the tradeoff between statistical detection rate, warning, and localization delay. In [31], the authors narrow down the classification of DDoS threats that support unusual behavior in the application layer and provide elliptical data on various DDoS tools. In addition, the author distinguishes methods of DDoS detective work based on viewing, blocking, detecting, and minimizing comments. In [32], a step-by-step approach to DDoS attack mitigation was presented, where the entire process of mitigating DDoS attacks was forced to a single layer or multiple layers. To increase the security of DDoS attacks, the go-layer process has become a useful solution. The authors of [33]

presented a new plastic strategy for detecting Al-DDoS attacks. Their aforementioned work differs from the previous method by considering the detection of Al-DDoS attacks in critical spine motions.

A distributed, useable, automated, and interactive ISP standard was presented by the study's authors in [34]. It not only distributes computing complexity and storage to adjacent places, but also facilitates the early identification of DDoS attacks and flash occurrences. Using an independent multi-agent system and agents that depend on particle evolution to facilitate effective communication and precise decision-making, the authors of [35] present a unique DDoS attack detection and prevention technique. Multiple intermediate agents are used to detect DDoS attacks, and the coordinating agent is updated. A secure root system and an access system that can identify nearby attacks on the RPL protocol have been suggested by experts of [36] in order to mitigate the effects of such attacks. To find the malicious node, the IDS is developed, taking into account the location data and the received signal strength. Researchers discovered perplexing real-time blocking DDoS application layer assaults on the web in [37] that seek to be discovered quickly and quickly. ARTP is a machine learning technique for quick and accurate app DDoS detection using multiple flood requests. The work's goal was achieved by measuring LLDoS databases through tests, and the findings showed how valuable the proposed model is.

A hybrid protocol proposed by the authors of [38] is the best suited protocol for cloud computing to detect DDoS attacks. The authors of [39] provide a new approach presented in this study. With the presence of these types of malicious nodes, attacks can be classified as active and inactive. In [40], the authors propose to identify the DDoS attack and mitigation model using the feature selection method. In the presented study, the network traffic is primarily analyzed according to the Hellinger degree. When a certain distance is detected, all data packets are analyzed and classified into two categories based on the selected segmentation factor, such as DDoS and official application groups [41]. The authors have addressed the problem and developed a secure system for these programs. The experts of [42] proposed a botnet detection method that can manage multiple datasets and also detect botnets in the network. In [43], researchers addressed the need to prevent DDoS attacks by defining and demonstrating a mixed identification model by introducing an advanced and effective method to identify and effectively distinguish flooding in a hot crowd.

In addition to introducing a multi-level classification method based on the presented set of entropy-based features with machine learning divisors to improve the low visibility and accuracy, the authors of [44] also introduced a set of novel entropy-based symbols to help reliably detect DDoS attacks. In [45], researchers discussed four important network protection schemes against end-to-end network attacks, end-to-end, victim, and distributed schemes with a focus on two innovative models, Gossip and D- WARD. In [46], the authors introduced a reduction method based on the fuzzy control system. It looks like inserting two new matrices.

In [47], the researchers presented a novel selection algorithm, Dynamic Ant Colony System, with a choice of three levels of renewal function. The presented method uses different levels of pheromones to make the ants stronger. The proposed method by the authors of [48] is contrasted with a different hybrid algorithm that is provided with 10-fold cross-validation. The proposed method outperforms existing methods in terms of accuracy, detection rate, and false alarm rate, according to the database-based test results of KDD CUP 99. In [49], the experts proposed a new method to mark a packet that could be forwarded from the attacker's side to the victim's side. It allows the victim to ensure the necessary protection for internet service providers (ISPs).

In the manuscript, [50], the authors propose a defense system called SkyShield. This scheme uses a graphical data framework to identify and mitigate DDoS attacks at the application level. First, they proposed new split calculations in two graphs that improve the effect of network dynamics and increase the accuracy of detection. Second, they used an atypical graph to help identify the malicious predators of a persistent attack. In [51],

the researchers proposed the concept of a system of experts. This program automatically resets security apps about incoming traffic. To achieve this, it is proposed to use a model, reasoning, and performance-based loop (LRA-M). In this case, it describes the structures of the corresponding system and defines its building blocks. In [52], the authors used a state-of-the-art SDN model, employing a new method for DDoS detection and mitigation known as State Sec. They demonstrated the benefits of this type of method, as shown in Figure 3.

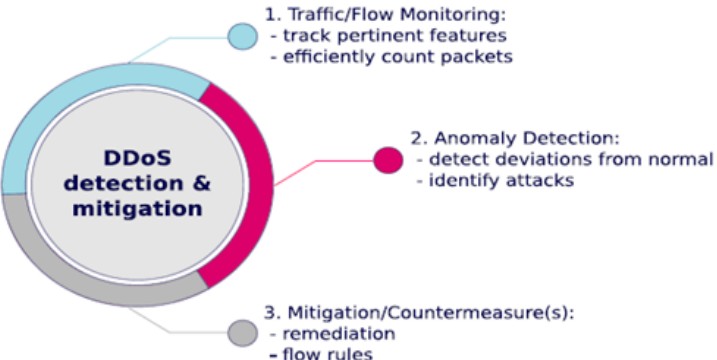

**Figure 3.** Key steps of DDoS detection and mitigation in software-defined networking.

The authors of [53] analyzed a number of current machine learning (ML) research projects that spread SDN for NIDS implementation. It was discovered that deep learning techniques had been looked into for the creation of SDN-based NIDS. Key steps of DDoS detection and mitigation in software-defined networking are shown in Figure 3. A brand-new authentication method was put forth in [53] as a defence against DDoS assaults on approved domain name servers. For duplicate resolvers, the solution employed the DNAME record to sign the domain redirection guide and then correctly reroute traffic to their downstream query domains. Many domains can be connected to vast and flexible provisioning and release of approved services to quickly raise demand in response to DDoS attacks. Results from imitations show that this solution works.

In [54], the authors proposed a new confusing discovery system with a unique parallel session feature attack detection (SFAD). The proposed process consists of two main steps. The first step is to set up smooth windows to collect web access information for different users; the second one is the PageRank method, used to control the weight information of web pages and calculate the similarity information for users. In [55], the authors proposed a IDS design that used ML algorithms such as Hidden Markov Model with a multi-pronged approach. This approach was developed and proven to solve common bugs in using the Hidden Markov Model in IDS, commonly referred to as the curse of size.

Based on psychologically inspired computations using entropy two-address representation, the inventors of the DDoS detection and defence technique developed their method in [56]. A vector segmentation technique is installed for support, the flow table features are retrieved, and the DDoS assault model is developed. In [57], the authors focused on internal DoS/DDoS attacks on WAMS devices using potential resources. To counter this type of attack, the authors propose an earlier and stronger extension of the multipath TCP (MPTCP) transport protocol, which they call MPTCP-H.

In [58], the authors proposed a fault detection method based on the study of mobile cloud computing that includes various client networks, as shown in Figure 4. The presented method does not require rule checking and its problem can be adapted to the needs of client networks. The authors of [59] have developed a collection-based approach to classify data representing the flow of network traffic. It combines normal traffic with DDoS.

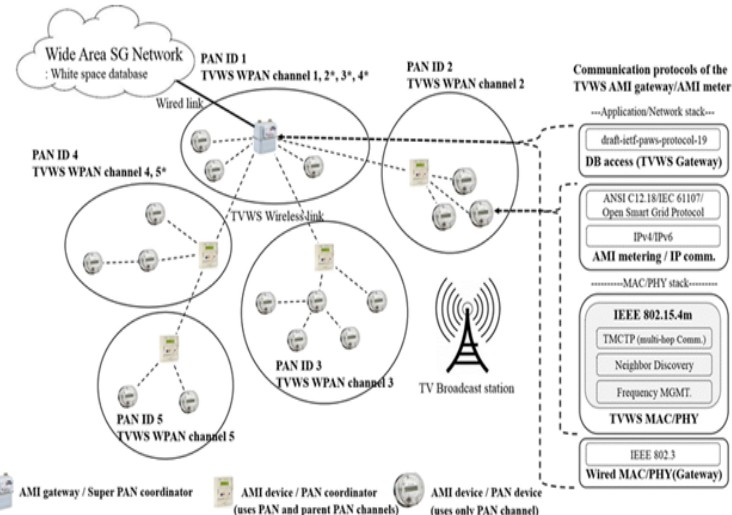

**Figure 4.** A fault detection method based on mobile cloud computing.

The authors of [60] provided a general overview of the use of SDN to improve network security, as shown in Figure 5. In particular, the authors examined recent research that emphasizes the use of SDN for network security. These include attack detection and mitigation, network traffic monitoring, service chaining, policy adaptation and management, centralized box deployment, and smart network security. On the basis of the newly announced Boltzmann Restricted Machines, the authors of [61] proposed an ingenious city-based diagnostic paradigm (RBMs). RBMs process the data produced by smart metres and sensors in real time by utilizing their capacity to unconventionally learn high-level aspects of raw data. In [62], the authors presented a new method for detecting the involvement of network devices in DDoS attacks. For this purpose, the traces next to the source are examined to detect inconsistent behavior.

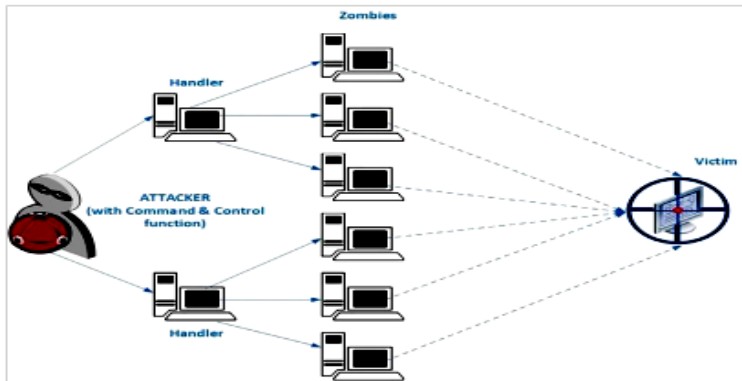

**Figure 5.** Common flow of DDoS attack.

In [63], the authors proposed ForChaos, a lightweight detection algorithm for IoT devices based on predictive and chaotic perception to detect flooding and DDoS attacks. In [64], the authors first developed the new Chinese Remainder Theorem based on the Reversible Sketch (CRT-RS). It can not only compress and consolidate large network traffic, but is also able to detect atypical keys as unwanted/malicious or network traffic sources. The literature review is summarized in Tables 1–3.

**Table 1.** Summary of literature review on DDoS attack detection.

| Paper | Year | Objective | Incremental Analysis | Assumptions Used | Relative Complexity | Real-Time Detection |
|---|---|---|---|---|---|---|
| [14] | 2017 | Mitigation of multi-page HTTP DDoS attacks | No | No | High | No |
| [15] | 2016 | To determine suitability of Levy walk for DDoS detection | No | Yes | Medium | No |
| [16] | 2017 | DDoS attack detection | Yes | Yes | Medium | No |
| [17] | 2017 | To detect DDoS attacks | No | Yes | High | No |
| [18] | 2017 | DDoS attack detection | No | No | High | No |
| [19] | 2016 | Attack detection | No | Yes | High | No |
| [20] | 2016 | Router protection system | No | Yes | Medium | No |
| [21] | 2016 | DDoS attack detection | No | Yes | Medium | No |
| [22] | 2017 | Increases resilience to DDoS attacks | No | Yes | High | No |
| [23] | 2016 | Flexible DDoS attack identifier | No | Yes | High | No |
| [24] | 2016 | DDoS attack detection | No | No | High | No |
| [25] | 2017 | Diagnosis method of attack on DoS applications | No | No | High | No |
| [26] | 2017 | Fast and timely location of a DDoS application over HTTP floods | No | No | Low | No |
| [27] | 2017 | Cloud-based firewall to reduce DDoS attacks | Yes | Yes | Medium | No |
| [28] | 2017 | To detect and prevent multiple DDoS TCP attack | No | No | High | No |
| [29] | 2017 | An event detection module | No | Yes | High | No |
| [30] | 2016 | Mine learning and DDoS base discovery | No | No | High | Yes |
| [31] | 2016 | Classification of DDoS threats | No | Yes | Medium | No |
| [32] | 2017 | DDoS attack mitigation | No | No | Medium | No |
| [33] | 2014 | Detection of Al-DDoS attacks | No | Yes | High | No |
| [34] | 2018 | Early identification of DDoS attacks and flash occurrences | No | Yes | Medium | No |

**Table 2.** Summary of literature review on DDoS attack detection.

| Paper | Year | Objective | Incremental Analysis | Assumptions Used | Relative Complexity | Real-Time Detection |
|-------|------|-----------|----------------------|------------------|---------------------|---------------------|
| [35] | 2018 | Unique DDoS attack detection and prevention technique | No | Yes | High | No |
| [36] | 2018 | Mitigation of the effects of DDoS attacks | No | Yes | High | No |
| [37] | 2018 | Real-time blocking DDoS application | No | Yes | High | Yes |
| [38] | 2018 | To detect DDoS attacks | No | No | Medium | No |
| [39] | 2018 | DDoS attacks classification as active and inactive | No | No | Medium | No |
| [40] | 2018 | Model to identify DDoS attack and mitigation | No | No | High | No |
| [41] | 2018 | Developed a secure system | No | No | High | No |
| [42] | 2018 | Manage multiple datasets and also detect botnets in the network | No | No | High | No |
| [43] | 2018 | To prevent DDoS attacks | No | No | High | No |
| [44] | 2018 | Novel entropy-based symbols to help reliably detect DDoS attacks | No | Yes | Medium | No |
| [45] | 2018 | Network protection schemes against end-to-end network attacks | No | No | High | No |
| [46] | 2018 | Reduction method based on the fuzzy control system | No | Yes | Medium | No |
| [47] | 2018 | Novel selection algorithm | No | No | High | No |
| [48] | 2018 | Accuracy of attack detection rate and false alarm rate | Yes | No | High | No |
| [49] | 2018 | To mark a malicious packet to be forwarded from the attacker | Yes | No | Medium | No |
| [50] | 2018 | Graphical data framework to identify and mitigate DDoS attacks | No | No | High | No |
| [51] | 2018 | Automatically resets security apps about incoming traffic | No | No | High | No |

**Table 3.** Summary of literature review on DDoS attack detection.

| Paper | Year | Objective | Incremental Analysis | Assumptions Used | Relative Complexity | Real-Time Detection |
|---|---|---|---|---|---|---|
| [52] | 2019 | DDoS detection and mitigation | Yes | No | High | No |
| [53] | 2019 | SDN-based NIDS | No | No | High | No |
| [54] | 2019 | Session feature attack detection | No | Yes | Medium | No |
| [55] | 2019 | To solve common bugs in IDS | No | No | Medium | No |
| [56] | 2019 | DDoS detection and defense technique | No | No | High | No |
| [57] | 2019 | To counter DoS/DDoS attacks on WAMS devices | No | No | High | No |
| [58] | 2019 | A fault detection method for mobile cloud computing | No | Yes | Medium | No |
| [59] | 2019 | Collection-based approach to classify the network traffic | No | Yes | Low | Yes |
| [60] | 2019 | To improve network security | Yes | Yes | Medium | No |
| [61] | 2019 | Ingenious city based diagnostic paradigm | No | Yes | High | No |
| [62] | 2019 | To detect malicious network devices involved in DDoS attacks | Yes | Yes | Medium | No |
| [63] | 2019 | A lightweight detection algorithm for IoT devices | No | Yes | Medium | No |
| [64] | 2019 | To detect malicious or network traffic sources | No | No | High | No |
| Ours | 2022 | Metrics-based DDoS attack detection | Yes | No | Low | Yes |

## 3. Motivation

In reviewing the above work, we have encountered a gap between ongoing research. The need for current and future technology is increasing. Various researchers have addressed DDoS attacks according to the literature review, but to some extent, they have only addressed one or two types of attacks and disregarded the rest. There are two major DDoS attacks that need to be addressed simultaneously, namely:

1. Increasing DDoS attack strategy;
2. Proxies DDoS attack strategy.

All this will help us in developing such algorithms that are secure enough so that they cannot be easily breached by attackers, and the unavailability of services may be avoided.

## 4. Proposed Research Methodology

Our proposed DDoS attack classification methodology (MLP classifier) algorithm is shown in Algorithm 1. The flowchart for the proposed system is shown in Figure 6. The main features of HTTP like GET and POST between other techniques, like TRACE, HEAD, DELETE, CONNECT, OPTIONS, and PUT, are studied. Ordinary legitimate clients have no more than 15–20 HTTP GET and POST demands per IP address. Bots become intelligent and imitate human behavior.

Usually, the same bots have the same HTTP request and spend the same time, same packet frame size and use increasing DDoS attack strategies to achieve their goal. Algorithm 1 captures the number of HTTP GET and POST request duration and packet size within 160 s.

The next feature states that the number of IP addresses is recorded and compared with the IP address list of the anonymous proxy server. The malicious user has launched multiple attacks through proxy servers to hide/backtrack his bots by performing a proxy DDoS attack, while a legitimate user mostly uses the real IP address to access the URL.

To gain access to the web server, the malicious user has employed several bots. Every cooperative bot opens a separate port whenever it reaches a specific destination port number, forwarding a large number of requests to the victim's web server. Long periods of time pass without any port connections being closed. A typical legitimate client opens ports, transmits data, and then shuts the connections. It was observed that the port numbers of authentic clients rarely varied. However, malevolent machines frequently have different port numbers that rise sequentially. The source port number's initial value was generated haphazardly. It has also been noted that the source port numbers in many DDoS attack tools begin with an arbitrary port number. Just one more digit than the one before it makes up the next port number. They establish a stable mapping to the destination server's port numbers.

The use of several bots to create a bot network is inherent in the head character of the bot or its master. Many vendor bots employ various manipulating codes to bring down a web server. It is well known that a client-side bot runs its code with a fixed log size and session duration.

## 5. Proposed Classification Model

This section contains an important discussion of our suggested model and the types of inputs that go into it. We combine four parameters into a mixture because no single parameter can clearly distinguish between a typical client or an assault from the dataset. These groupings are examined to determine whether a specific group value places a given IP address in the category of attacks, suspicious, or regular clients. We have grouped the four parameters according to the following values:

1. 1st parameter (TQ value);
2. 2nd parameter (PKQ value);
3. 3rd parameter (PX-IP value);
4. 4th parameter (SourcePort value).

---

**Algorithm 1** Pseudo-code for DDoS attack detection

---

 1: S-IP ← Source IP
 2: ProxyIP-count ← initialize
 3: **if** S-IP == GET_request or S-IP == POST_request **then**
 4: Go to step 7
 5: **else**Go to step 1
 6: **end if**
 7: Se_Time [ ] = SourceIP_Se_Time
 8: Packet_lenght [ ] = SourceIP_paketlength
 9: **if** S-P == Anonymous_ProxyList-IP **then**
10: ProxyIP-count++
11: **else**Go to step 13
12: **end if**
13: **if** S_IP-Port==Constant **then**
14: SourcePort == Constant
15: **else**SourcePort == Varying
16: **end if**
17: **for** i in range (0, Se_Time[ ]) **do**
18:     **for** j in range (i + 1, Se_Time[ ]) **do**
19:         **if** arr[i] == arr[j] **then**
20: No of host equal Session Time ++
21:         **end if**
22:     **end for**
23: **end for**
24: **for** i in range (0, Time_Frame[ ]) **do**
25:     **for** j in range (i + 1, Packet_length[ ]) **do**
26:         **if** arr[i] == arr[j] **then**
27: No of host equal Packet frame Size ++
28:         **end if**
29:         **if** No of host equal Session Time $\geq$ 20 **then**
30: TQ == High
31:             **else**TQ == Low
32:             **end if**
33:         **if** No of host equal Packet Frame size $\geq$ 20 **then**
34: PKQ == High
35:             **else**PKQ == Low
36:             **end if**
37:         **if** ProxyIP-count $\geq$ 20 **then**
38: PX-IP == High
39:             **else**PX-IP == Low
40:             **end if**
41:     **end for**
42: **end for**

---

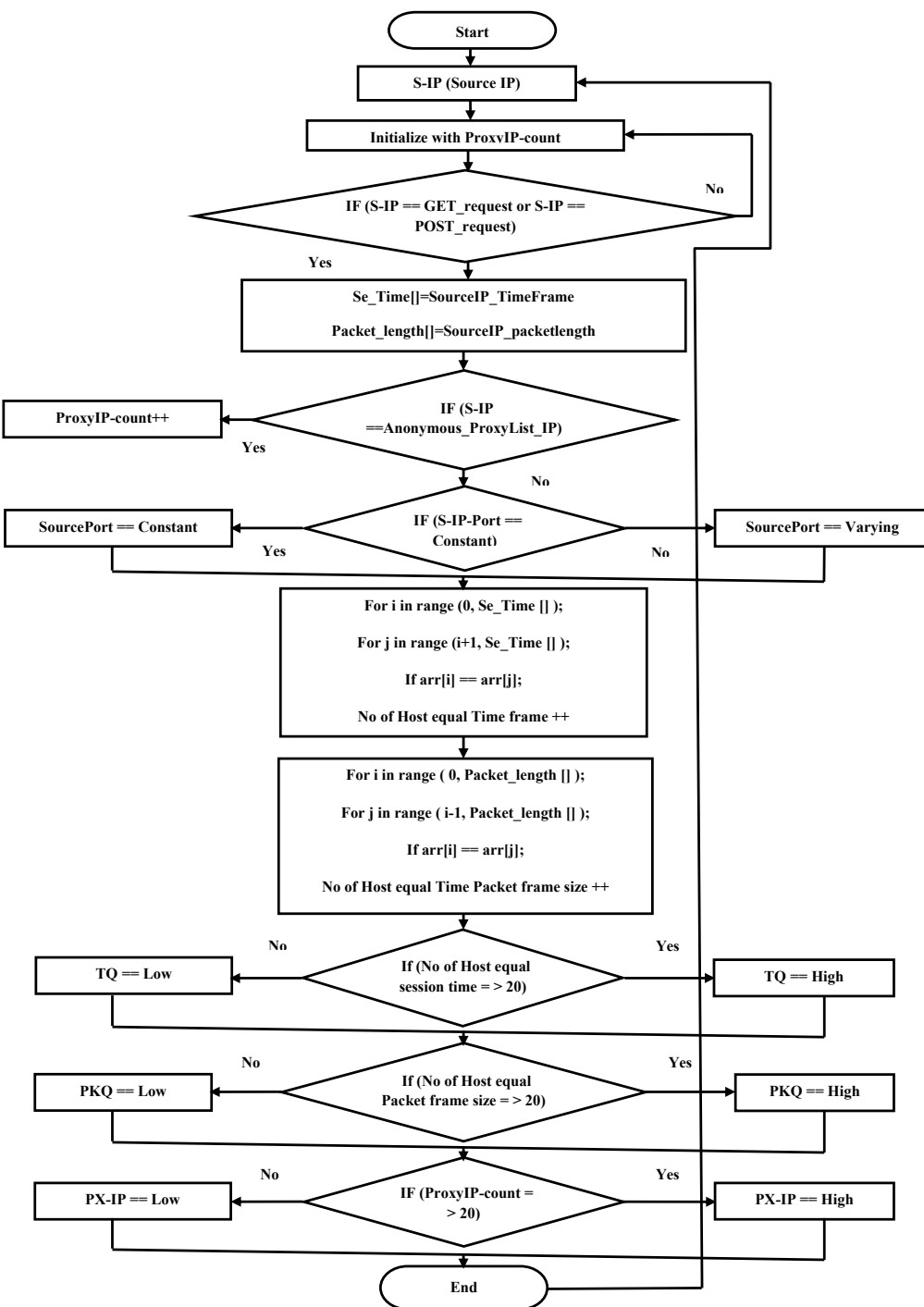

**Figure 6.** Flow chart of the proposed methodology.

Our proposed algorithm uses high and low values for the first, second, and third parameters and varying and fixed values for the fourth parameter, as shown in Table 4. Figure 7 depicts the structure of the chosen network. The error histogram of the chosen network is shown in Figure 8.

**Table 4.** Grouping of different parameters for DDoS attack detection criteria. Note: TQ: no. of host spent equal session time; PKQ: no. of host equal packet length; PX-IP: no. of host use proxy server; SourcePort: host changing ports; PD: pattern detection.

| No | TQ | PKQ | PX-IP | SourcePort | PD |
|----|------|------|--------|-------------|------------|
| 1 | Low <=20 | Low <=20 | Low <=20 | Constant <=20 | Normal |
| 2 | Low | Low | Low | Varying | Suspicious |
| 3 | Low | Low | High | Constant | Normal |
| 4 | Low | Low | High | Varying | Suspicious |
| 5 | Low | High | Low | Constant | Normal |
| 6 | Low | High | Low | Varying | Suspicious |
| 7 | Low | High | High | Constant | Suspicious |
| 8 | Low | High | High | Varying | Attack |
| 9 | High | Low | Low | Constant | Normal |
| 10 | High | Low | Low | Varying | Suspicious |
| 11 | High | Low | High | Constant | Suspicious |
| 12 | High | Low | High | Varying | Suspicious |
| 13 | High | High | Low | Constant | Attack |
| 14 | High | High | Low | Varying | Attack |
| 15 | High | High | High | Constant | Attack |
| 16 | High | High | High | Varying | Attack |

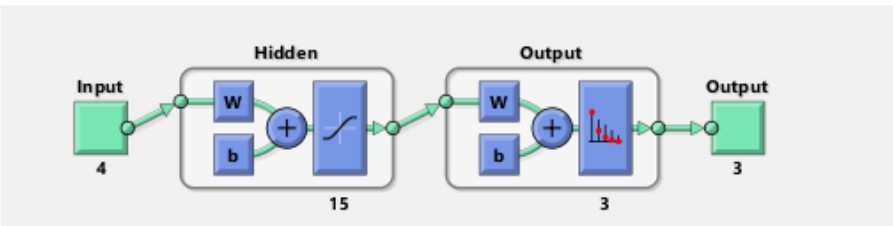

**Figure 7.** Structure of the chosen network.

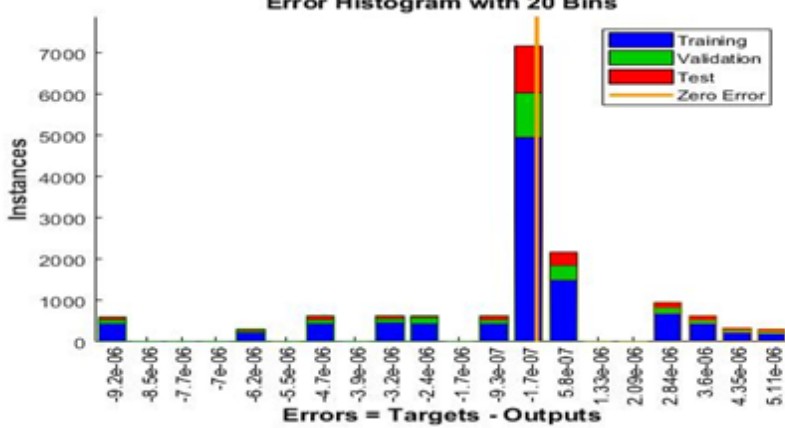

**Figure 8.** Error histogram of the chosen network.

Figure 9 shows a schematic diagram of an artificial neural network (ANN). Further study regarding different variants of neural networks can be seen in [65]. Equation (1) is used to calculate the derivative of error with respect to (w.r.t.) output weights ($w_{jk}$).

$$\frac{\partial E}{\partial w_{jk}} = \delta_k a_j \tag{1}$$

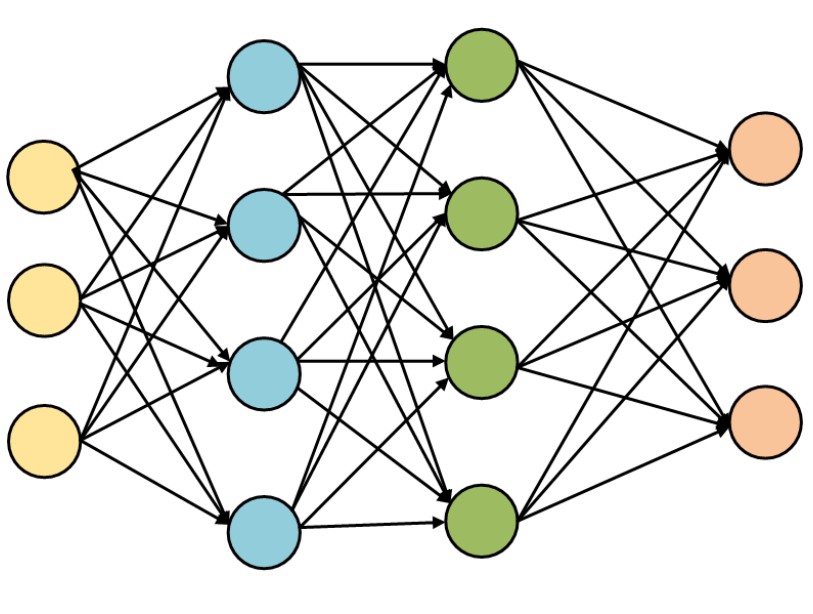

**Input layer**     **Hidden layer 1**     **Hidden layer 2**     **Output layer**

**Figure 9.** A schematic representation of an artificial neural network.

The following Equation (2), is used for calculation of error derivatives w.r.t. output layer biases ($b_k$).

$$\frac{\partial E}{\partial b_k} = a_k - t_k g_{k'} z_k \frac{\partial E}{\partial b_k} = \delta_k \tag{2}$$

The weights of hidden layers are calculated using following Equation (3).

$$\frac{\partial E}{\partial w_{ij}} = \sum_{k \in K} (a_k - t_k) g_{k'}(z_k) w_{jk} g_{k'}(z_j) a_i$$
$$\frac{\partial E}{\partial w_{ij}} = g_{j'}(z_j) a_i \sum_{k \in K} (a_k - t_k) g_{k'}(z_k) w_{jk} \tag{3}$$
$$\frac{\partial E}{\partial w_{ij}} = a_i g_{j'}(z_j) \sum_{k \in K} (\delta_k) w_{jk}$$

## 6. Simulation Setup, Evaluation Criteria, Dataset, Results and Discussion

### 6.1. Simulation Setup

For this study, the web log of the network traffic accessing the Apache web server is recorded using pcap. We also used the web log to create a dataset that contains a mixture of the four parameters. We examine the MLP classifier using a test as an example. Using Equations (1)–(3), we obtain an estimate for the connection weights in the hidden and output layers. In this study, we calculate criteria such as accuracy, false positive rate (FPR), and false negative rate (FNR). These standards are evaluated for models with eleven classifications to obtain a perfect difference of performances.

*6.2. Evaluation Criteria*

6.2.1. Accuracy

One of the evaluation criteria that determines the model's overall accuracy is accuracy. The percentage of all samples that the classifier successfully classifies is known as overall accuracy. Equation (4) can be used to calculate accuracy.

$$Accuracy = \frac{TP + TN}{TP + TN + FP + FN} \tag{4}$$

When an observation is true positive (TP), it means that it is both predicted to be positive and is, in fact, positive. A true negative (TN) observation is one that is both predicted to be and really is negative. False positive (FP) observations are those that were expected to be positive, but turned out to be negative. False negative (FN) observations are those that are projected to be negative, but turn out to be positive.

6.2.2. Receiver Operating Characteristic (ROC) Curve

ROC curve is a graph that represents the performance of a classification model at each classification threshold. It represents two metrics, namely the true-positive rate (TRP) and the false-positive rate (FPR). The TPR or sensitivity is a proxy for the recovery rate and is therefore expressed in Equation (5).

$$TPR = \frac{TP}{TP + FN} \tag{5}$$

FPR or specificity can be expressed with the help of Equation (6).

$$FPR = \frac{FP}{FP + TN} \tag{6}$$

The ROC is used to plot TPR against FPR at different classification thresholds. At a lower classification threshold, more elements are classified as positive, increasing both the FPR and TPR.

*6.3. Dataset*

We have divided the whole dataset into three data sets, namely, a verification set (15%), a test set (15%), and a training set (70%). The verification set is used to measure the network efficiency and stop training upon normal stopping criteria. The test set does not affect training, and thus, provides an independent measure of network performance. The training set is used for training of our proposed model. In the training phase, the best verification performance occurs during the 37th round, and the network is configured during the 37th round. Evaluation of test input, shown in Table 5, provides the corresponding values for the specified errors.

**Table 5.** Evaluation of test input.

| Parameter(s) | Train X | Y | Validation X | Y | Test X | Y | Best X | Y |
|---|---|---|---|---|---|---|---|---|
| Min | 0 | $2.969 \times 10^{-7}$ | 0 | $2.856 \times 10^{-7}$ | 0 | $2.78 \times 10^{-7}$ | 0 | $9 \times 10^{-8}$ |
| Max | 40 | 0.712 | 40 | 0.6377 | 40 | 0.6668 | 40 | 1.1 |
| Mean | 20 | 0.03123 | 20 | 0.02871 | 20 | 0.02995 | NaN | NaN |
| Median | 20 | $7.524 \times 10^{-5}$ | 20 | $7.166 \times 10^{-5}$ | 20 | $7.222 \times 10^{-5}$ | NaN | NaN |
| Mode | 0 | $2.969 \times 10^{-7}$ | 0 | $2.856 \times 10^{-7}$ | 0 | $2.78 \times 10^{-7}$ | 40 | $2.856 \times 10^{-7}$ |
| Std | 11.98 | 0.1157 | 11.98 | 0.1044 | 11.98 | 0.1092 | NaN | NaN |
| Range | 40 | 0.712 | 40 | 0.6377 | 40 | 0.6668 | 40 | 1.1 |

Figure 10 shows the performance diagram. From the performance diagram, the user can see the current status of the training process. The X-axis of this diagram indicates the number of iterations, while the Y-axis indicates the cross-entropy value for each iteration. The blue line graph represents the training results, the green represents the validation results, and the red represents the testing results. This performance graph is calculated for each iteration in the training process. The graph in which all three results of training, validation, and testing match in almost all points is selected as the best performance. The best performance value is $2.9778 \times 10^{-7}$, which means that the behavior of the network is stable, and its generalizability is sufficiently high. Evaluation results of Data1 and Data2 are shown in Table 6. Evaluation results of training, testing, and validation of the dataset are shown in Table 7.

**Table 6.** Evaluation results of Data1 and Data2.

| Parameter(s) | Data1 | | Data2 | |
|---|---|---|---|---|
| | X | Y | X | Y |
| Min | 0 | $9.768 \times 10^{-7}$ | 0 | 0 |
| Max | 40 | 0.3977 | 40 | 0 |
| Mean | 20 | 0.03192 | 20 | 0 |
| Median | 20 | 0.0003166 | 20 | 0 |
| Mode | 0 | $9.768 \times 10^{-7}$ | 0 | 0 |
| Std | 11.98 | 0.08324 | 11.98 | 0 |
| Range | 40 | 0.3977 | 40 | 0 |

**Table 7.** Evaluation results of training, testing, and validation of dataset.

| Parameter(s) | Training | | Validation | | Test | | Zero Error | |
|---|---|---|---|---|---|---|---|---|
| | X | Y | X | Y | X | Y | X | Y |
| Min | $-1.037 \times 10^{-5}$ | 0 | $-1.037 \times 10^{-5}$ | 0 | $-1.037 \times 10^{-5}$ | 0 | 0 | 0 |
| Max | $4.449 \times 10^{-6}$ | 5031 | $4.449 \times 10^{-6}$ | 6142 | $4.449 \times 10^{-6}$ | 7208 | 0 | 7929 |
| Mean | $-2.959 \times 10^{-6}$ | 525 | $-2.959 \times 10^{-6}$ | 637.5 | $-2.959 \times 10^{-6}$ | 750 | 0 | 3964 |
| Median | $-2.959 \times 10^{-6}$ | 219 | $-2.959 \times 10^{-6}$ | 262 | $-2.959 \times 10^{-6}$ | 305 | 0 | 3964 |
| Mode | $-1.037 \times 10^{-5}$ | 0 | $-1.037 \times 10^{-5}$ | 0 | $-1.037 \times 10^{-5}$ | 0 | 0 | 0 |
| Std | $4.614 \times 10^{-6}$ | 1107 | $4.614 \times 10^{-6}$ | 1352 | $4.614 \times 10^{-6}$ | 1587 | 0 | 5607 |
| Range | $1.482 \times 10^{-5}$ | 5031 | $1.482 \times 10^{-5}$ | 6142 | $1.482 \times 10^{-5}$ | 7208 | 0 | 7929 |

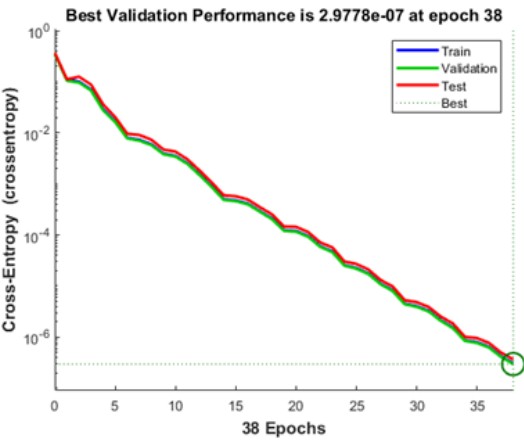

**Figure 10.** Performance plot of the chosen network.

### 6.4. Results and Discussion

Using Equations (4)–(6), we performed the comparison of different classification models by means of ROC (receiver operating characteristics) curve using Weka tool version 3.9.3. The performance confusion matrix of the proposed MLP classification model is shown in Figure 11 and ROC curves of proposed MLP classification model is depicted in Figure 12. A clear comparison of different classifiers with our proposed MLP classifier is shown in Figure 13a–i. We calculated the accuracy of our proposed MLP classification model with Naïve Bayes, Decision Stump, Logistic Model Tree, Naïve Bayes Updateable, Naïve Bayes Multinomial Text, AdaBoostM1, Attribute Selected Classifier, Iterative Classifier Optimizer, and OneR, as shown in Figure 14.

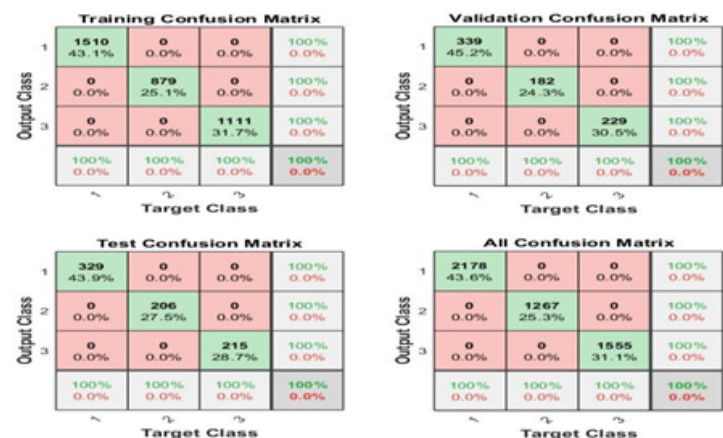

**Figure 11.** Performance confusion matrix of proposed MLP classification model.

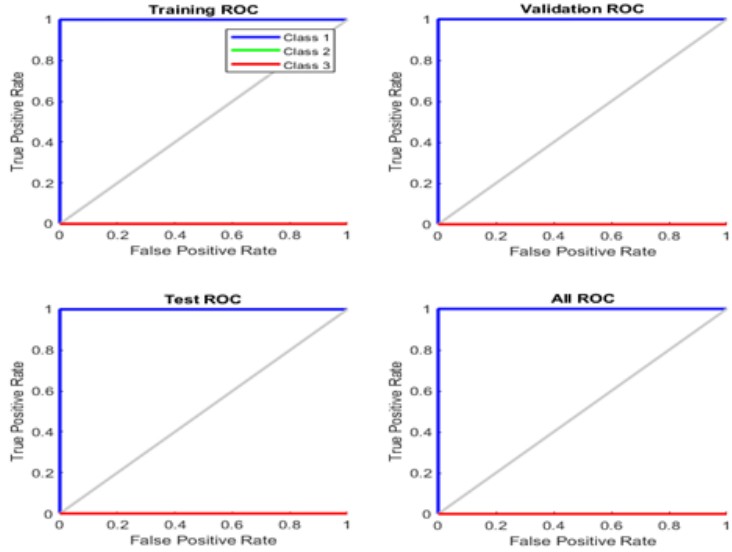

**Figure 12.** Receiver operating characteristic (ROC) curves of proposed MLP classification model.

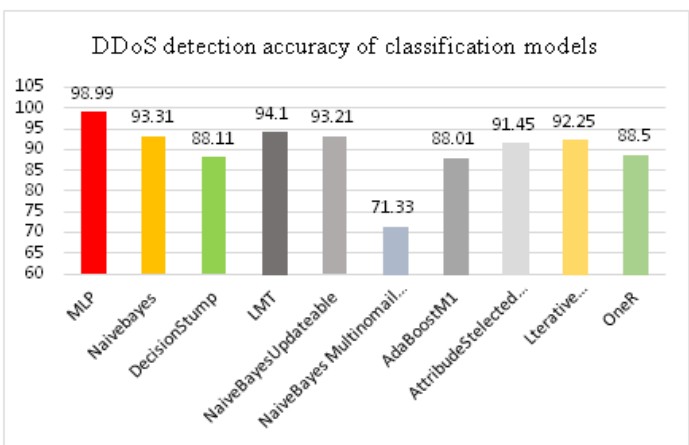

(**a**) MLP vs. Naïve Bayes.

(**b**) MLP vs. Decision Stump.

(**c**) MLP vs. LMT.

(**d**) MLP vs. Naive Bayes Updateable.

(**e**) MLP vs. Naïve Bayes Multinomial Text.

(**f**) MLP vs. AdaBoostM1.

(**g**) MLP vs. Attribute Selected Classifier.

(**h**) MLP vs. Iterative Classifier Optimizer.

(**i**) MLP vs. OneR.

**Figure 13.** ROC curves showing comparison of proposed MLP classifier with counterparts.

**Figure 14.** Accuracy comparison of proposed MLP classifier with other classification models.

It is clear from the results shown in Table 8 that MLP classifier outperforms all other classification models with an efficiency of 98.99%. Using our proposed MLP classifier, we can quickly recognise DDoS assaults at the application level. We are able to identify between legitimate clients and attackers using the proposed MLP classifier. Some of the presumed IP addresses, meanwhile, do not fit the mould of a typical client or an attacker. In this study, we tested the effectiveness of our proposed technique by using it to identify attacks in real-world DDoS attack datasets including CTU-13 (2011), our company's web logs (2019), and our own dataset. The detection accuracy analysis of ten classifiers is shown in Figure 14.

**Table 8.** Comparison of proposed model (MLP) with other models. Note: NB: NaiveBayes, DS: DecisionStump, LMT: Logistic Model Tree, NBU: NaiveBayesUpdateable, NBMT: NaiveBayes Multinomial Text, ASC: Attribute Selected Classifier, ABM1: AdaBoostM1, ICO: Iterative Classifier Optimizer, MLP: Multilayer Perceptron, CM: Confusion Matrix

| Criteria | NB | DS | LMT | NBU | NBMT | ABM1 | ASC | ICO | OneR | MLP (Proposed) |
|---|---|---|---|---|---|---|---|---|---|---|
| CM $\begin{bmatrix} a & b \\ c & d \end{bmatrix}$ | $\begin{bmatrix} 420 & 45 \\ 22 & 10 \end{bmatrix}$ | $\begin{bmatrix} 400 & 40 \\ 35 & 200 \end{bmatrix}$ | $\begin{bmatrix} 435 & 33 \\ 40 & 225 \end{bmatrix}$ | $\begin{bmatrix} 410 & 44 \\ 25 & 234 \end{bmatrix}$ | $\begin{bmatrix} 314 & 29 \\ 26 & 233 \end{bmatrix}$ | $\begin{bmatrix} 425 & 34 \\ 21 & 212 \end{bmatrix}$ | $\begin{bmatrix} 408 & 22 \\ 15 & 211 \end{bmatrix}$ | $\begin{bmatrix} 419 & 41 \\ 27 & 219 \end{bmatrix}$ | $\begin{bmatrix} 425 & 44 \\ 27 & 217 \end{bmatrix}$ | $\begin{bmatrix} 489 & 50 \\ 30 & 40 \end{bmatrix}$ |
| Accuracy | 0.9331 | 0.8811 | 0.9401 | 0.9323 | 0.7133 | 0.8801 | 0.9145 | 0.9254 | 0.8835 | 0.9899 |
| TP | 0.9631 | 0.8901 | 0.9400 | 0.9000 | 0.6900 | 0.8611 | 0.8900 | 0.8800 | 0.8455 | 0.9799 |
| FP | 0.432 | 0.5670 | 0.0996 | 0.0690 | 0.9998 | 0.9897 | 0.8675 | 0.7989 | 0.9698 | 0.0211 |

## 7. Conclusions and Future Work

This study proposes a MLP classification model for internal data to identify DDoS attacks at the application level. According to this research work, features from the incoming network traffic are considered, which have large differences in their characteristics. In this research work, all possible groupings of attack features were found and a decree was structured to distinguish an attacker, a suspect, or an authentic client. The research results show that our proposed MLP classification model provides 98.99% accuracy in identifying DDoS attacks at the application level by FP of 2.11%. In the future, we plan to address the problem of improving the DDoS attack detection accuracy. We will extend this research to distinguish application-level DDoS attacks from flash events by studying the different access behaviors. Furthermore, we will work to investigate the possibility and feasibility of implementation of our proposed MLP DDoS attack classification methodology to a real-time cyberattack detection system.

Future work could focus on providing an application or service that can quickly analyze and benchmark each new dataset using algorithms selected by the researcher. The application would be able to answer the question of which dataset performs better with which algorithms. This would be a great help to researchers who are looking for a high performing dataset and also want a consistent approach to results by using the best performing datasets and algorithms. Computational complexity will also be calculated and addressed in the future extension of the current research.

**Author Contributions:** Methodology, S.A. (Sheeraz Ahmed), Z.A.K. and S.M.M.; Software, S.A. (Sheeraz Ahmed), Z.A.K., S.M.M. and S.L.; Validation, S.M.M., S.A. (Sheeraz Aslam) and H.M.; Formal analysis, S.A. (Sheeraz Aslam), S.M.M., M.A. and H.M.; Investigation, S.A. (Sheeraz Ahmed), M.A. and Z.N.; Data curation, S.A. (Sheeraz Ahmed), M.A., S.L. and Z.N.; Writing—original draft, S.A. (Sheeraz Ahmed), Z.A.K. and S.M.M.; Writing—review & editing, S.A. (Sheeraz Aslam), H.M., S.L. and Z.N.; Visualization, S.A. (Sheeraz Aslam), Z.A.K. and Z.N.; Supervision, S.A., (Sheeraz Ahmed) Z.A.K. and S.M.M.; Project administration, S.A. (Sheeraz Ahmed), S.M.M., S.L., S.A. (Sheeraz Aslam) and H.M.; Funding acquisition, S.M.M., S.A. (Sheeraz Aslam) and H.M. All authors have read and agreed to the published version of the manuscript.

**Funding:** This research received no external funding.

**Data Availability Statement:** Not applicable.

**Conflicts of Interest:** The authors declare no known conflict of interest.

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
