# Peer review of "Effective and Efficient DDoS Attack Detection Using Deep Learning Algorithm, Multi-Layer Perceptron"

_futureinternet, doi:10.3390/fi15020076_

Round 1

Reviewer 1 Report

Comments for Authors

The authors proposed MLP classification model for internal data to identify DDoS attacks at the application level. They addressed the DDoS attacks by analyzing the characteristics of incoming packets, including the size of HTTP frame packets, the number of Internet Protocol (IP) addresses sent, constant mappings of ports, and the number of IP addresses using proxy IP. The deep learning algorithm has been used to evaluate the metrics-based attack detection effectiveness. My comments to the authors:

1) The title is "Reducing the Impact of Distributed Denial of Service Attacks on the Application Layer" which is totally different from the contribution of the authors, DDoS attack classification. So, authors must change the title to fit the proposed algorithm.

2) The authors have a shortage in the networking knowledge: the ports addresses (transport layer), IP addresses (network layer) and the SYN floods (TCP in transport layer) as in Figure 2 are not in the application layer. You have to delete the specific word "application layer" and replaced it with TCP/IP layers.

3) Subsections 1.1, and 1.2 are explained some thing else than that in Figures which means you must rewrite them to explain the correct information in the Figures 2, and 3.

4) Write the problem statement or few sentences that can explain the challenges that are going to solve before the contribution paragraph.

5) Most of the Figures are not clear. So, I could not read them.

6) Where is the machine learning (deep learning algorithm) in your Algorithm 1.

7) Do you think that the title of Algorithm 1 " Pseudo-code for route discovery " is related to the paper or some thing else. This is because route discovery is special for routing protocol.

Author Response

Dear Reviewer,

Thank you very much for your time and efforts to review our paper. I believe that the quality of our manuscript has been enhanced after considering your comments/ suggestion. please see the attached file for detailed responses to your comments.

Thanks,

BR,

Sheraz

Reviewer 2 Report

Dear Authors

The paper titled “Reducing the Impact of Distributed Denial of Service Attacks on the Application Layer” This study addressed novel application-layer DDoS attacks by analyzing the characteristics of incoming packets, including the size of HTTP frame packets, the number of Internet Protocol (IP) addresses sent, constant mappings of ports, and the number of IP addresses using proxy IP. A deep learning algorithm, a multilayer perceptron (MLP), is used to evaluate the metrics-based attack detection effectiveness.

The paper is interesting however it needs improvements.

1.      Extensive English editing is required throughout the manuscript.

2.      First three paragraphs in the introduction section are supported by only 2 works from 2017. Authors should add [1, 2].

3.      Line 77 needs proper reference.

4.      The Figures are very blur, all figures needs to be of good resolution and label should be readable.

5.      Related Works should first describe the ML and DL in general for other applications to justify why these are used in the current manuscript [3].

6.      Overfitting and model tuning is required, see and add CDLSTM and SMOTEDNN.

7.      # of params are required for the models with FLOPS and the computational complexiety.

8. Limitations and the future scope should be added with more clarity.

9. Experiment environment with computational complexity should be added.

10. Authors need to provide the merits of this study vs. other review studies.

11. The inter-comparison or comparison with other studies is missing, please add them.

 References

 1.     Development of PCCNN-Based Network Intrusion Detection System for EDGE Computing

2.     DNNBoT: Deep Neural Network-Based Botnet Detection and Classification;

3.     Insider Threat Detection Based on NLP Word Embedding and Machine Learning

Author Response

(The authors gave the same response as above.)

Round 2

Reviewer 1 Report

The authors addressed all my comments.

Reviewer 2 Report

Dear Authors

I have now completed the review of the revised manuscript, titled " Effective and Efficient DDoS Attack Detection using Deep Learning Algorithm, Multi-Layer Perceptron”. I have observed that the authors put in good efforts to address most of the comments satisfactorily. Best wishes